# Migrant Workers and Psychological Health: A Systematic Review

**Nicola Mucci [1],\*, Veronica Traversini [1], Gabriele Giorgi [2], Eleonora Tommasi [1], Simone De Sio [3] and Giulio Arcangeli [1]**

[1] Department of Experimental and Clinical Medicine, University of Florence, 23 Largo Piero Palagi, I-50139 Florence, Italy; veronica.traversini@unifi.it (V.T.); eleonora.tommasi@unifi.it (E.T.); giulio.arcangeli@unifi.it (G.A.)

[2] Department of Human Sciences, European University of Rome, 190 Via degli Aldobrandeschi, I-00163 Rome, Italy; gabriele.giorgi@unier.it

[3] Department of Anatomical, Histological, Forensic and Locomotor Apparatus Sciences, Sapienza University of Rome, 5 Piazzale Aldo Moro, I-00185 Rome, Italy; simone.desio@uniroma1.it

\* Correspondence: nicola.mucci@unifi.it; Tel.: +39-055-417-769

**Abstract:** Migrant workers show an increase in the incidence of serious, psychotic, anxiety, and post-traumatic disorders due to a series of socio-environmental variables, such as loss of social status, discrimination, and separations from the family. The purpose is to elaborate a systematic review and highlight the prevailing psychological pathologies of these workers and categories most at risk. Our research included articles published from 2009 to 2019 on the major databases (Pub Med, Cochrane Library, and Scopus) using a combination of some keywords. The online search indicated 1.228 references. Using inclusion and exclusion criteria, we analyzed 127 articles, in particular 12 reviews and 115 original articles. Principal emerging disorders from the research are depressive syndrome (poor concentration at work, feeling down, or anger and somatization), anxiety, alcohol or substance abuse, and poor sleep quality. This causes low life conditions, which is also due to marginalization from the social context and strenuous work; in fact, migrant workers may suffer verbal or physical abuse, and they are often employed in dangerous, unhealthy jobs. It is therefore essential to increase the role of occupational medicine and promote wellbeing for this vulnerable job category.

**Keywords:** migrant workers; mental health; psychological disorders; systematic review; organizational psychology; occupational health

## 1. Introduction

In the last decades, Europe has become the destination of growing migration flows from middle and low-income countries. According to the most recent estimates provided by the United Nations Organization, there are 232 million international migrants in the world (3.2% of the total population worldwide), a number that has progressively increase compared to 1990 (154 million) and 2000 (175 million).

Of these, 58.6% were resident in developed countries, while the remaining 96 million (41.4%) came from developing countries. About two thirds of the entire international migrant population is concentrated in Europe and Asia. In 2013, 72 million international migrants lived in Europe (31.0%), and 71 million lived in Asia (30.6%). Women, with a 48% presence, constitute an increasingly significant reality within the phenomenon [1].

In recent years, migratory flows have undergone variations, moving toward regions able to offer more job opportunities. In fact, since 2013, over 51% of international migrants have been concentrating

in countries such as United States (19.8%), Russian Federation (4.7%), Germany (4.3%), and Saudi Arabia (3.9%).

According to the estimates reported in the XXIII ISMU Report on Migration 2017, there are about 5.5 million legal immigrants in Italy, and the number of foreigners employed in 2016 reached 2,401,000, representing 10.5% of total employment [2].

A series of interacting socio-environmental variables can seriously compromise mental health characterizes migration [3]. In the scientific literature, migrants show more psychotic, mood, anxiety disorders and post-traumatic disorders [4]. The risk increases for first-generation migrants coming from low-income countries because geographical and cultural distances from the original country and the transition to urban areas is crucial in determining conditions of psychophysical stress. The most important risk factors include failure to complete projects, cross-cultural stress, and loss of social status, discrimination, and separations from the family context [5]. The presence of compatriots, on the other hand, is a protective factor, even though it can sometimes hinder integration into the community, threatening long-term adaptation.

Moreover, immigrants in Europe are particularly disadvantaged in accessing care and mental health services due to social, linguistic, cultural, and organizational barriers [6,7].

Regarding employment, immigrant workers are more often employed in strenuous, precarious jobs, without protections and social protection networks, in a labor market that, due to the economic crisis, has increased competition and risks of discrimination [8,9]. Discrimination is a potential determinant of problems of psychological distress, anxiety, and depressive symptoms that, in turn, can result in isolation and addiction to drugs and/or alcohol [10].

A recent study in Italy confirmed this trend, pointing out that 15.8% of immigrants in Italy report that they were discriminated against in the workplace because they were foreigners. The study showed a greater probability psychological distress in those who declared that they had suffered discrimination on the job because they were foreigners, arrived in Italy for at least five years, and who came from the South American countries [11].

Globalization, changes and accelerated rhythms characterize the 21th century and business world has been affected by insecurity, instability, economic crisis [12]. For this reasons, individual well-being and work organization can be compromised [13,14]; however, experts in the field can apply methods of the psychology of sustainability for the protection and development of vulnerable workers. In particular, harmonization is a new approach, because it considers geographical and temporal factors, with a look to the future, using thoughtful, individual and community processes. [15,16].

*Aim*

The purpose of this research is to carry out a systematic review of the literature relating to mental health of migrant workers, at the international level, in order to identify the main disorders manifested by this vulnerable working category, the most risky occupational sectors, the most involved geographical areas, and, finally, highlighting some suitable preventive strategies that have emerged from the research.

## 2. Materials and Methods

### 2.1. Literature Research

The research included articles published from 2009 to 15 March 2019, on the major online databases (Pubmed, Cochrane Library, and Scopus). The search strategy used a combination of controlled vocabulary and free text terms based on the following keywords: migrant workers, mental health, diseases, illnesses, travelers, work, occupational medicine, depression, and work-related stress. All research fields were considered. Additionally, we practiced a hand search on reference lists of the selected articles and reviews to carry out a wider analysis.

Two independent reviewers read titles and abstracts of the reports identified by the search strategy. They selected relevant reports according to inclusion and exclusion criteria. Doubts or disagreements

were solved by discussion with a third researcher. Subsequently, they independently screened the corresponding full text to decide on final eligibility. Finally, the authors eliminated duplicate studies and articles without full texts.

### 2.2. Quality Assessment

Three different reviewers assessed the methodological quality of the selected studies with specific rating tools. We used the International Narrative Systematic Assessment (INSA) method [17] to judge the quality of narrative reviews, AMSTAR to evaluate systematic reviews, and the Newcastle Ottawa Scale to evaluate cross-sectional, cohort studies, and case control studies [18], while the JADAD scale was applied for randomized clinical trials [19].

### 2.3. Eligibility and Inclusion Criteria

No restrictions were applied for language or publication type. The studies included in this review focus on the mental health of migrant workers and the main diseases found in this particular category of workers.

### 2.4. Exclusion Criteria

We have excluded reports not related to psychological disturbances of migrant workers, articles that did not speak of workers, articles related to physical disorders or publications about children. We have also excluded reports of less academic significance, editorial, individual contributions and descriptive studies published in scientific conferences without any quantitative and qualitative inferences.

### 3. Results

The online research indicated 1228 references from PubMed (992), Scopus (205), and Cochrane Library (31). Of these, 872 were excluded because not related to the psychological disorders of migrant workers. Of the remaining articles, 160 were excluded because doubles.

Finally, 127 studies were included in the analysis (Figure 1).

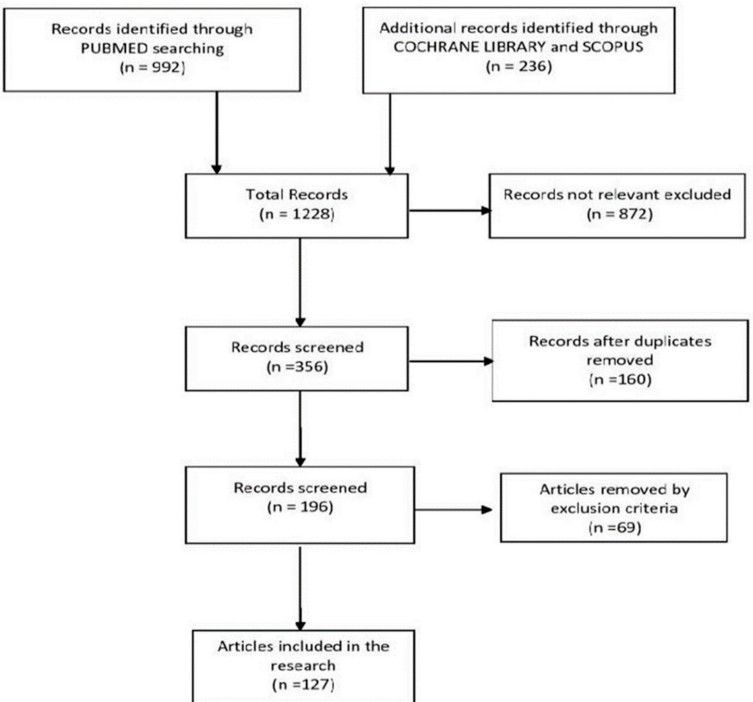

**Figure 1.** Flow-chart of the bibliographic research.

Of these, nine are systematic reviews, three are narrative reviews, and 115 are original articles. In reference to the latter, 108 are cross-sectional studies, 3 trials, 2 cohort studies and 2 case-control studies (Figure 2).

China and the US are countries in which the most studies have been published (43 and 21 articles, respectively). The most articles were published in 2016 (19 studies), followed by 2017 and 2018 (18 articles), signifying ever increasing attention to the problems concerning migrant mental health. The selected articles mainly investigate the anxious depressive symptoms shown by migrants (44 studies; 34.3%) and work-related stress (31 articles; 24.2%).

Employees in factories, construction sites, and services are the prevalent examined job categories (36 articles; 28.1%), followed by farmworkers and domestic workers (28 and 13 studies; 21.8% and 10.1%, respectively).

| Author | Year | Country | Type of study | Author | Year | Country | Type of study |
| --- | --- | --- | --- | --- | --- | --- | --- |
| Akhter et al | 2017 | Bangladesh | cross sectional | Meyer SR et al | 2014 | Thailand | cross sectional |
| Al-Maskari F et al | 2011 | Arabia | cross sectional | Mora DC et al | 2016 | USA | cross sectional |
| Ang JW et al | 2017 | Singapore | cross sectional | Mou J et al | 2013 | China | cross sectional |
| Anjara SG et al | 2017 | Singapore | cross sectional | Mou J et al | 2013 | China | systematic review |
| Arcury TA et al | 2016 | USA | cross sectional | Nadim W et al | 2016 | Saudi Arabia | cross sectional |
| Ayalon L | 2012 | Israel | cross sectional | Negi NJ | 2013 | USA | cross sectional |
| Ayalon L | 2009 | Israel | cross sectional | Nguyen et al | 2016 | Vietnam | cross sectional |
| Bacio G et al | 2014 | USA | cross sectional | Nilvarangkul K et al | 2010 | Thailand | cross sectional |
| Bousman CA et al | 2011 | Mexico | cross sectional | Palupi et al | 2017 | Taiwan | cross sectional |
| Burke S et al | 2012 | USA | cross sectional | Pocock et al | 2016 | Thailand | cross sectional |
| Bustamante et al | 2017 | Brazil | narrative review | Pocock et al | 2018 | Thailand | systematic review |
| Capasso R et al | 2018 | Italy | cross sectional | Qiu P et al | 2011 | China | cross sectional |
| Cayuela A et al | 2015 | Spain | cross sectional | Ramos AK et al | 2015 | USA | cross sectional |
| Chaney BH et al | 2017 | USA | cross sectional | Ramos AK et al | 2016 | USA | cross sectional |
| Chen G et al | 2015 | China | cross sectional | Reid A | 2012 | New Zealand | cross sectional |
| Chen Li et al | 2012 | China | cross sectional | Reus-Pons M et al | 2018 | SW Europe | case control |
| Cheung JTK et al | 2019 | China | cross sectional | Robert G et al | 2014 | Spain | cross sectional |
| Cieslik A | 2011 | Uk | cross sectional | Roblyer MI et al | 2016 | USA | cross sectional |
| Costa PFFD et al | 2017 | Brazil | cross sectional | Rodriguez G et al | 2016 | USA | cross sectional |
| Crawford JO et al | 2011 | Uk | systematic review | Rosano A et al | 2012 | Italy, Spain | cross sectional |
| Cui X et al | 2012 | China | cross sectional | Sandberg et al | 2012 | USA | cross sectional |
| Daly A et al | 2019 | Australia | cross sectional | Sandberg et al | 2014 | USA | cross sectional |
| Dick M et al | 2015 | Israel | cross sectional | Sandberg et al | 2016 | USA | cross sectional |
| Dunlavy AC et al | 2013 | Sweden | cross sectional | Sandstrom E et al | 2015 | Sweden | cross sectional |
| Espinoza Castro et al | 2018 | Germany | cross sectional | Schilgen B et al | 2019 | Germany | cross sectional |
| Fernandes C et al | 2016 | Portugal | systematic review | Schilgen B et al | 2017 | Germany | systematic review |
| Gao J et al | 2014 | China | cross sectional | Simsek Z et al | 2016 | Turkey | cross sectional |
| Garabiles MR et al | 2019 | China | cross sectional | Smith LH et al | 2013 | Denmark | trial |
| Georges A et al | 2013 | USA | cross sectional | Sterud et al | 2018 | Norway | systematic review |
| Gilbert L et al | 2019 | Kazakhstan | cross sectional | Steward et al | 2018 | India | cross sectional |
| Green O et al | 2018 | Israel | cross sectional | Subedi RP et al | 2016 | Canada | cross sectional |
| Grzywacz JG et al | 2011 | USA | cross sectional | Subedi RP et al | 2014 | Canada | cross sectional |
| Grzywacz JG et al | 2010 | USA | cross sectional | Sweileh WM | 2018 | Palestine | systematic review |
| Grzywacz JG et al | 2009 | China | cross sectional | Tarricone I et al | 2017 | Italy | cohort study |
| Habtamu et al | 2017 | Africa | cross sectional | Teixeira et al | 2018 | Portugal | cross sectional |
| Hall BJ et al | 2018 | China | cross sectional | Tsay SY | 2012 | Taiwan | cross sectional |
| Hall BJ et al | 2019 | China | cross sectional | Vahabi M et al | 2017 | Canada | cross sectional |
| Hall BJ et al | 2019 | China | cross sectional | Van der ham et al | 2015 | Netherlands | cross sectional |
| Hall BJ et al | 2019 | China | cross sectional | Vanhoutte et al | 2018 | Australia | cross sectional |
| Han L et al | 2014 | China | cross sectional | Viveros-Guzman A et al | 2015 | Canada | cross sectional |
| Heeren M et al | 2014 | Switzerland | cross sectional | Wang Z et al | 2016 | China | cross sectional |
| Holumyong C et al | 2018 | Thailand | cross sectional | Warfa N et al | 2012 | UK, USA | cross sectional |
| Hsieh YC et al | 2016 | USA | cross sectional | Weine S et al | 2012 | Russia | cross sectional |
| Ismayilova L et al | 2014 | Central Asia | cross sectional | Winkelman SB et al | 2013 | USA | cross sectional |
| Kaewanuchit C | 2016 | Thailand | cross sectional | Wu J et al | 2011 | China | cross sectional |
| Kronfol et al | 2014 | Qatar | narrative review | Xiao Y et al | 2018 | China | cross sectional |
| Kumparatana P et al | 2017 | Kazakhstan | cross sectional | Yang H et al | 2017 | China | cross sectional |
| Lau JT et al | 2012 | China | narrative review | Yang H et al | 2015 | China | cross sectional |
| Lee H et al | 2012 | Korea | cross sectional | Yang T et al | 2012 | China | cross sectional |
| Li J et al | 2017 | China | systematic review | Yao C et al | 2015 | New Zealand | cross sectional |
| Li J et al | 2014 | China | cross sectional | Yu B et al | 2017 | China | cross sectional |
| Li W et al | 2018 | China | cross sectional | Zahreddine N et al | 2014 | Lebanon | case control |
| Lin QH et al | 2012 | China | cross sectional | Zaller N et al | 2014 | China | cross sectional |
| Liu Y et al | 2015 | China | cross sectional | Zeng Z et al | 2014 | China | cross sectional |
| Loret de Mola C et al | 2012 | Perù | cross sectional | Zhang L et al | 2015 | China | cross sectional |
| Lu Y | 2010 | Indonesia | cohort study | Zhong BI et al | 2018 | China | cross sectional |
| Luo H et al | 2016 | China | cross sectional | Zhong BI et al | 2017 | China | cross sectional |
| Luo M et al | 2018 | China | cross sectional | Zhong BI et al | 2015 | China | cross sectional |
| Malhotra R et al | 2013 | USA | systematic review | Zhong BI et al | 2016 | China | cross sectional |
| McCoy et al | 2016 | USA | trial | Zhong BI et al | 2018 | China | cross sectional |
| Mendoza NB et al | 2017 | China | cross sectional | Zhu C et al | 2012 | China | cross sectional |
| Meyer SR et al | 2016 | Thailand | cross sectional | Zhu C et al | 2013 | China | trial |
|  |  |  |  | Zhu Y et al | 2017 | China | cross sectional |

**Figure 2.** Manuscripts included in the review, in alphabetical order.

### 3.1. Reviews

As regards quality, the AMSTAR score shows an average of 6.8, a median of 7, and modal values of 7 and 8, thus indicating an intermediate level (Figure 3). Best systematic reviews come from Portugal and Thailand (AMSTAR = 8). As regards narrative reviews scores, INSA score shows an average 7.5 and a median of 5.

Fernandes and colleagues found that the mental health of migrant workers and their psychosocial well-being deteriorated regardless of the psychosocial factor assessed. This deterioration turns into an increase in physiological and cognitive stress, resulting in specific conditions, like back pain, shoulder pain, headaches, gastrointestinal disorders, ischemic heart disease, diabetes, mental illness and suicides. Deteriorated psychosocial environment appears to be associated with higher levels of absenteeism in the workplace. Gender differences were evident in the perception of stress symptoms; in fact, women reported higher levels of stress symptoms compared to male colleagues with similar tasks [20].

| Author | Type of article | Prevalent disease | Type of workers | Score |
|---|---|---|---|---|
| McCoy | trial | alcohol abuse | farmworkers | J.3 |
| Smith LH | trial | work-related stress | employees in cleaning industries | J.2 |
| Zhu C | trial | job satisfaction | employees in factories | J.3 |
| | | | | |
| Bustamante | narrative review | PTSD | not specified | I.6 |
| Crawford JO | systematic review | psychological distress | manager, engineers, technicians | A.6 |
| Fernandes C | systematic review | work-related stress | employees, health care workers | A.8 |
| Kronfol | narrative review | depression and suicidal behaviors | housemaids | I.5 |
| Lau JT | narrative review | suicides | employees in factories | I.4 |
| Li J | systematic review | mental well-being | not specified | A.6 |
| Malhotra R | systematic review | psychotic and mood disorders | domestics | A.7 |
| Mou J | systematic review | work-related stress | not specified | A.5 |
| Pocock | systematic review | psychological distress (depression, anxiety and anger) | fishers and seafarers | A.8 |
| Schilgen B | systematic review | mental well-being | nurses | A.7 |
| Sterud | systematic review | psychological distress | various (factories, health care, manual workers) | A.8 |
| Sweileh WM | systematic review | psychological distress | farmworkers | A.7 |
| | | | | |
| Reus-Pons M | case control | depression | employees | N.6 |
| Zahreddine N | case control | work-related abuse and psychosis | domestics | N.5 |
| Lu Y | cohort study | work-related stress and depression | employees | N.7 |
| Tarricone I | cohort study | psychotic disorders | employees | N.7 |

**Figure 3.** Review, trial, case-control and cohort studies with specific diseases, job categories and relative scores.

Schilgen studied the mental condition of nurses, finding that adaptation to a foreign country, lifestyles and poor language proficient can lead to stress. Foreign nurses who reported being discriminated against in the workplace tended to be more dissatisfied. In particular, Filipino-American nurses have reported more anxiety, anger and sadness than their Caucasian colleagues because they report daily discrimination, racism and bullying at work, as unequal career advancements, unequal pay, insufficient training [21].

In Crawford's review, two articles examined work and mental health factors; in one, poor mental health was significantly associated with high demands, low decision-making authority, low skill discretion, role conflict, insecurity, unclear roles. The other reports that many musculoskeletal symptoms are linked to psychosocial factors, particularly cervical pain is linked to higher job demands, low control overtime and high levels of competition among colleagues. While shoulder-localized symptoms were associated with high work demands and high levels of uncertainty, as well as low back pain at low levels of interaction with colleagues [22].

Mou's review found that rural-urban Chinese migrants are suffering from work stress due to economic pressure, workload, distance from family members and discrimination. Several studies on the psychological outcomes of Chinese migrants have symptoms related to major depression and insomnia; this probably contributes to high suicide rates or suicide attempts by this category [23].

Pocock examined a particular category of migrants: Cambodian fishermen. Violence and isolation at sea for long periods aggravate mental problems. Anger, anxiety, stress, memory loss, aggression and

substance abuse were observed among returning anglers, with higher rates of suicidal thoughts and attempted suicide [24].

Sterud's study found that the risk of mental disorders was higher among people from ethnic minorities who reported receiving unfair treatment or racial slurs. However, studies in Scandinavia, Spain and Arab countries point out that difference in working conditions have a negligible impact on the increased risk of poor mental health among immigrants [25].

Factory workers in China often increase their income by working overtime. Lau and colleagues found that high work demands combined with a lack of autonomy in decision-making can cause stress; work is often very repetitive, mechanical and without regular training courses, all elements that do not help migrant workers to successfully pursue a new life [26].

The literature examined by Sweileh and colleagues included a relatively good volume of publications on the mental health of migrant workers. The main factors leading to higher levels of stress among these workers are a difficult adaptation to the new conditions, as well as barriers to access to health services and physical abuse. For this reason, the risk of depression, stress and suicide among migrant workers is higher. [27].

Kronfol analyzed the literature on the migrant workers' mental health in Arabia, highlighting a greater risk of developing depression and suicidal tendencies. He found that domestics are a vulnerable category, often subjected to abuse and harassment [28].

Similarly, Malhotra and colleagues identify maids as a particularly at risk working category, putting in evidence adverse work conditions and associated health problems, such as physical, verbal, and sexual abuse at the workplace, caregiving tasks associated with musculoskeletal strain, and poor mental health with psychotic, neurotic, and mood disorders [29].

Bustamante points out that migration is associated with specific stressors, mainly related to the experience and acculturation process. These important stressors have potential consequences in many areas, including mental health. The prevalence of post-traumatic stress disorder among migrants is very high (47%), especially among refugees [30].

Li's review examines evidence of rural–urban migrants' mental health status and its association with various dimensions of social exclusion. The authors found conflicting evidence on the mental health status of migrants in comparison with non-migrants, but they did find strong evidence that social exclusion is negatively associated with migrants' mental health. Limited access to full labor rights and experience of social stigma, discrimination, and inequity were the most significant factors [31].

*3.2. Original Articles*

The values assigned to the original articles have an average of 6.7, a median and a modal of 7 (Figure 4). This situation amount to a discrete quality of the studies; American, Chinese, and Swiss articles obtained the highest values (NEW CASTLE = 8).

In order to carry out the results and considered the quantity of the selected articles, we proceed with a subdivision based on the principal migrant workers' problems found by the authors.

| Author | Prevalent Disease | Work Categories | Score |
|---|---|---|---|
| Aalto AM | work-related stress | physicians | N.7 |
| Agudelo-Suarez | poorer mental health | not specified | N.7 |
| Akhter | work-related stress and anxiety | employees in industries | N.6 |
| Al-Maskari F | depression and suicidal behaviors | construction | N.7 |
| Ang JW | psychological distress (depression and anxiety) | employees in construction sites | N.7 |
| Anjara SG | work-related stress | domestics | N.7 |
| Arcury TA | alcohol abuse | farmworkers | N.6 |
| Ayalon L | depression, suicidal behaviors, abuse | home care workers | N.7 |
| Ayalon L | work-related abuse, injuries and burnout | home care workers | N.7 |
| Bacio G | depression and alcohol abuse | day labourers (farmworkers, construction etc.) | N.8 |
| Bousman CA | neuropsychological impairments | farmworkers and homemakers | N.6 |
| Burke S | prepardness and barrier in accessing services | farmworkers | N.7 |
| Capasso R | work-related stress and job satisfaction | employees in factories and costruction | N.7 |
| Cayuela A | poorer mental health | manual workers | N.7 |
| Chaney BH | depression and stress | farmworkers | N.7 |
| Chen G | PTSD | employees | N.6 |
| Chen L | depression | employees | N.7 |
| Cheung JTK | abuse and depression | domestics | N.7 |
| Cieslik A | work-related stress and job satisfaction | employees | N.4 |
| Costa PFFD | depression, anxiety and fatigue | sugarcane workers | N.6 |
| Cui X | work-related stress and tobacco dependence | employees (factory, construction, service, transport) | N.7 |
| Daly A | poorer mental health | various (professional, trade, service,sales,clericals) | N.7 |
| Dick M | emotional distress | employees | N.7 |
| Dunlavy AC | poorer mental health | employees | N.7 |
| Espinoza Castro | distress and violence | employees | N.7 |
| Gao J | alcohol abuse | employees | N.7 |
| Garabiles MR | depression and anxiety | domestics | N.7 |
| Georges A | depression | farmworkers | N.7 |
| Gilbert L | depression and alcohol abuse | vendors | N.7 |
| Green O | work-related abuse | home care workers | N.7 |
| Grzywacz JG | depression and sleepiness | farmworkers | N.7 |
| Grzywacz JG | depression | farmworkers | N.6 |
| Grzywacz JG | work-related stress | farmworkers | N.5 |
| Habtamu | fatigue, sleepiness, suicidal ideation, depression | farmworkers and employees | N.7 |
| Hall BJ | depression and anxiety | domestics | N.7 |
| Hall BJ | depression, anxiety and rumination | domestics | N.7 |
| Hall BJ | depression and anxiety | domestics | N.6 |
| Hall BJ | depression, anxiety and alcohol abuse | domestics | N.6 |
| Han L | work-related stress | employees in factories | N.7 |
| Heeren M | depression and anxiety | employees | N.8 |
| Holumyong C | acculturative stress and poorer mental wellbeing | farmworkers, fishermen, construction, factories, domestic | N.7 |
| Hsieh YC | work-related stress | Hotel housekeepers | N.5 |
| Ismayilova L | depression and alcohol abuse | employees in bazaars and markets | N.7 |
| Kaewanuchit C | poor mental wellbeing | employees | N.7 |
| Kumparatana P | depression and alcohol abuse | vendors | N.7 |
| Lee H | work-related stress and depression | employees (services, construction and factories) | N.7 |
| Li J | poor mental wellbeing | employees in service o business sector | N.7 |
| Li W | work-related stress and depression | employees and farmworkers | N.7 |
| Lin QH | psychological distress | employees | N.7 |
| Liu Y | tobacco dependence | employees in construction, restaurant, factories | N.7 |
| Loret de Mola C | depression, anxiety and somatic distress | not specified | N.7 |
| Luo H | work-related stress and burnout | employees in factories | N.7 |
| Luo M | suicidal ideation | employees | N.7 |
| Mendoza NB | depression and anxiety | domestics | N.7 |
| Meyer SR | depression and anxiety | farmworkers | N.6 |
| Meyer SR | depression and anxiety | farmworkers, employees in factories and sex industries | N.7 |
| Mora DC | depression and alcohol abuse | farmworkers | N.8 |
| Mou J | tobacco dependence | employees in factory | N.7 |
| Nadim W | depression | employees in a company | N.7 |
| Negi NJ | psychological distress (discrimination, isolation) | employees in construction | N.7 |
| Nguyen | dissatisfaction and work-related stress | employees in factories | N.5 |
| Nilvarangkul K | work-related stress | employees | N.5 |
| Palupi | depression and fatigue | domestics | N.7 |
| Pocock | depression, anxiety and suicidal thoughts | fishers and seafarers | N.5 |
| Qiu P | depression | employees | N.7 |
| Ramos AK | depression | farmworkers | N.7 |
| Ramos AK | depression and stress | farmworkers | N.7 |
| Reid A | poorer mental health | various (employees, bachelor, trade..) | N.7 |
| Robert G | poor mental wellbeing | employees in industries and agriculture | N.7 |
| Roblyer MI | depression | farmworkers | N.7 |
| Rodriguez G | work-related stress | employees in service and production | N.6 |
| Rosano A | work-related stress | employees (industries, services, construction) | N.7 |
| Sandberg | depression and sleepiness | farmworkers | N.7 |
| Sandberg | poor sleep quality | farmworkers | N.7 |
| Sandberg | sleep disorders | farmworkers | N.7 |
| Sandstrom E | poorer mental health | employees | N.5 |
| Schilgen B | work-related stress | home care nurses | N.6 |
| Simsek Z | work-related stress | farmworkers | N.7 |
| Steward | alcohol abuse | employees in construction, sales, factories, bank | N.7 |
| Subedi RP | work-related stress | employees | N.7 |
| Subedi RP | work-related stress and job satisfaction | taxi drivers, store workers, gas station workers | N.7 |
| Teixeira | psychological distress | employees (services, construction) | N.7 |
| Tsay SY | work-related stress | white-collar | N.7 |
| Vahabi M | work-related stress | caregivers | N.5 |
| Van der ham | work-related stress | domestics | N.7 |
| Vanhoutte | work stress and depression | employees | N.7 |
| Viveros-Guzman A | workplace miscommunications and tensions | farmworkers | N.6 |
| Wang Z | anxiety, mood disorders and substance abuse | not specified | N.8 |
| Warfa N | depression and suicidal behaviors | various (employees, driver, physician etc) | N.6 |
| Weine S | PTSD | employees in bazaars and construction sites | N.7 |
| Winkelman SB | work-related stress and depression | farmworkers | N.5 |
| Wu J | nicotine dependence | employees (factory, construction, service) | N.7 |
| Xiao Y | depression and unhappiness | employees | N.7 |
| Yang H | somatization, obsessions-compulsions, depression | employees (factory, construction, service, domestics) | N.7 |
| Yang H | poor mental wellbeing | employees in construction, factories, service | N.7 |
| Yang T | work-related stress | employees (machinery, transport) | N.7 |
| Yao C | work-related stress | employees (agriculture) | N.7 |
| Yu B | depression and anxiety | employees | N.7 |
| Zaller N | alcohol abuse | sex industries | N.7 |
| Zeng Z | worse psychological well-being | employees | N.7 |
| Zhang L | worse psychological well-being | employees | N.7 |
| Zhong BI | psychological distress (depression and anxiety) | employees | N.7 |
| Zhong BI | suicidal ideation | employees | N.7 |
| Zhong BI | poor mental wellbeing | employees in factories | N.7 |
| Zhong BI | depression | employees in factories | N.7 |
| Zhong BI | work-related stress | employees in factories | N.5 |
| Zhu C | work-related stress and job satisfaction | manual workers | N.7 |
| Zhu Y | schizophrenia | not specified | N.7 |

**Figure 4.** Cross-sectional studies with specific diseases, job categories, and relative scores, in alphabetical order.

### 3.2.1. Depression and Anxiety

Several authors have analyzed the mental health status of farmers, especially Latino emigrants in the US, and domestic workers.

In Meyer's study, for male farmers, depressive symptoms are more associated with workplace insecurity ($p < 0.01$), while coercions or aggressions are mostly related to anxious states (p-0.0); for female workers, the factors implicated in the genesis of depression are daily problems and safety violations ($p > 0.01$). [32].

Also, Ramos examined the agricultural sector. In this research, 200 migrant farm workers (mean age 33.5 years, 93% men, 92% Mexican, 59% low education) were interviewed. The results indicate that work-accidents are the triggering factor; in fact, workers with a positive history of trauma report a 7 times higher risk than those without injury [33]. In 2015, Ramos himself sought to identify the main stressors involved in developing depressive symptoms among farmers using the Migrant Farm

Worker Stress Inventory (MFWSI). 45.8% of respondents were depressed. The main correlations were found in some domains, in particular economy, logistics, integrity (r—0.22, $p < 0.01$) [34].

In the study carried out by Grzywacz in 2010, there was no evidence that depressive symptoms differed from whether or not agricultural workers had an H2A visa, by worker status (migrant versus seasonal) or by a level of acculturation measured by the ability to understand English. Positive correlations, however, were found with the marital status; singles had more frequent depressive symptoms than married ones. [35].

Georges and colleagues examined the association between elevated depressive symptoms and healthcare utilization among 2905 Latino farm workers interviewed for the National Agricultural Workers Survey; 11% farm workers had elevated depressive symptoms, predominantly men (75%), over 35 years of age, with little formal education. Farm workers with elevated depressive symptoms had significantly higher odds of utilizing health care in the US [36].

Costa carried out a cross-sectional study with 110 sugarcane workers who work in the region of São José do Rio Preto. Most of them were between 18 and 29 years of age (n.69), single (n.53), childless (n.59), and elementary schooled (n.83). The prevalence of mental disorders was higher among workers who were older than 50 years (60%), married (47.4%), who did not have children (40.7%), with one to five years of sugarcane and who consumed alcoholic beverages (41.2%). "Feeling dissatisfied" received more responses (72.7%), followed by "feeling nervous, tense or worried" (59.1%). Among the somatizations, the poor appetite prevails 66%. To be noted that among the group with obvious signs of deflected mood, 3.6% experienced suicidal thoughts [37].

Hall BJ examined domestics' mental health in four research studies conducted in 2018 and 2019. Participants were recruited using snowball sampling methods through a local nongovernmental organization (NGO) in Macao, China; the sample was mainly composed of women, around 40 years old, married and they reported attending some college or graduating from college.

In the first study, the author found that stress and burnout were associated with the following symptoms: the inability to concentrate on job, difficulty sleeping, feeling down or angry, and heart palpitations. Depression-like syndrome was noted, which included any signs (gloomy face, looking haggard, always sad, social withdrawal, laziness, inability to do tasks, becoming slow, and thinking of suicide) [38,39]. In subsequent articles analyzing Filipina domestic workers in China, the authors found that discrimination was significantly associated with depression (rs = 0.43) and anxiety (rs = 0.42), while social capital was significantly negatively associated with depression ($\beta = -0.30$) and anxiety ($\beta = -0.32$). PCL-5 scores correlated strongly with scores on measures of depression ($\varrho = 0.71$, $p < 0.001$), anxiety ($p < 0.001$), and rumination ($p < 0.001$) [34,36]. Finally, domestic workers reported physical (e.g., hypertension, chronic pain, diabetes, poor sleep), mental health problems (depression, anxiety), addictive behaviors (gambling, alcohol misuse) [40,41].

In addition, Mendoza's study evaluated the role of migration stressors on poor mental health among Filipino female domestic workers. The author found that post migration stress and social network support from friends were associated with greater depressive symptom severity ($p < 0.01$) and anxiety symptoms. In fact, participants with greater post-migration stress reported higher anxiety symptoms and somatization symptoms if they were exposed to greater social network support from friends. [42].

Zhong, in 2017, found that psychological distress was more prevalent in the newer rather than older generation migrant workers (36.2% vs. 28.2%). For the newer generation, main risk factors included low monthly income, recent two-week physical morbidity, migrating alone, poor Mandarin proficiency and long working hours, while for the older generation's correlates included low education, recent two-week physical morbidity and having worked in many cities [43].

Nadim, in the Al-Qassim region of Saudi Arabia, conducted a cross-sectional survey of 400 migrant workers in a selected company, which only employ men. The inclusion criteria included Non-Saudi and working in KSA for at least six months. The participants' mean age was 36.5 years; the majority were from South Asia, and Indian was the largest nationality (43%). More than three-fourths of the

sample had education up to high-school, and a large majority of them were married. Nadim found that depression prevalence was 20%; it varies based on age, stress and self-reported health [44].

Among 1180 migrant workers studied by Qiu and recruited as part of the program on health demands in China, which was conducted from September 2008 to July 2009, 23.7% had clinically relevant depression symptoms (CES-D score ≥ 16), and 12.8% were consistent with a clinical diagnosis of depression (CES-D score ≥ 21). Self-rated economic status, city adaptation status, and self-rated health had negative effects on depression [45].

Also, discrimination and social isolation are risk factors for the development of depression. In Roblyer's analysis, most participants (84.7%) were aged 18–35 years, were married or living as married (90.3%), had been born in Mexico (85.9%), and had been living in the United States for at least five years (91%). Nearly three-quarters of the women (74.3%) had at most nine years of formal education, and slightly fewer (72.6%) lived in a non-migrant (i.e., seasonal) farmworker household. Participants' average depressive symptoms score was 7.2 (SD = 5.6), and nearly one-third of women (31%) met the threshold level of depressive symptoms. Correlational analyses demonstrated that depressive symptoms were positively associated with family conflict and perceived racial or ethnic discrimination [46]. Xiao compared migrant workers with locals; migrants were exposed to a poorer residential environment. In fact, these poorer housing conditions were the result of residential discrimination. Consequently, there were also some differences in mental health between migrants and locals. "Unhappy and depressed" scores for migrants (at the alpha = 0.05 level) were significantly lower than those of locals, implying that there is a higher prevalence of unhappiness and depression among migrants [47].

In addition, Negi and colleagues examined 150 Latino laborers, to correlate discrimination and social isolation with psychological distress. Participants ranged in age from 18 to 68 years old, generally reported lower levels of acculturation and less than a high school education. The majority were married (59%), with their wives living in their country of origin. The correlation analysis revealed that psychological distress was related positively with discrimination ($p < 0.01$) and social isolation ($p < 0.01$). Acculturation was negatively associated with percent of income sent home ($p < 0.05$) and social isolation ($p < 0.05$), while this last was positively associated with discrimination ($p < 0.01$) [48].

Lu Yao confirm this hypothesis; in this research, labor migrants were much more likely to report depressive symptoms than non-migrants were ($p < 0.001$), presumably due to family separation and reduced social support. The detrimental impact of migration on mental health existed only for solo migrants ($p < 0.001$), but not for migrants accompanied by family members. They also compared rural–urban migrants with urban non-migrants, which showed a much smaller difference between the two groups; this was largely due to the higher prevalence of depressive symptoms in urban areas than in rural areas, because urban living usually associated with more adverse social environment and greater awareness of mental well-being [49].

The rural migrant workers, examined by Loret De Mola and colleagues, confirm this theory. In their study, 90% had migrated to Lima more than 20 years ago, the majority were older than 12 years at the time of migration, and approximately 50% of the group had spent at least half of their lifetime in the urban area. The overall prevalence of common mental disorders was 39.4% in this group who moved to the city, often with lower levels of education and lower family income [50].

Many authors highlight how the female gender is more vulnerable and tends to show anxiety-depressive symptoms. Of these, only 12.7% of Chaney's sample had high scores, mostly women (74%), with 4–10 years of residence (47.3%), married (31.6%), without health insurance (100%), within the monthly income bracket of 500-700 Dollars (47.3%), and educated at the level primary school (47.3%). In addition, 11.3% indicated depressive symptoms; of these, the majority are women (64.7%), residents between 11 and 20 years (52.9%), free marriage (47%) primary school education (53%) [51].

In Palupi's study, Indonesian women migrants who working in Taipei were studied, 99.4% ($n = 193$) of whom were Muslim, the mean age was 33.0 ± 7.0 years, and the mean BMI was 23.6 ± 3.4 kg/m$^2$. The prevalence of their fatigue was 27.8%; depressive symptoms ($p < 0.0001$), cooking

methods ($p = 0.027$), and self-perceived feelings of sadness and weakness ($p = 0.003$) were associated with fatigue [52].

Sandstrom analyzed women migrants from Yugoslavia in Sweden; they experienced that physical activity contributed positively to their mental health, but at the same time, their mental health problems could act as barriers against physical activity sometimes. They had experienced increased concentration, happiness, and improvements in mood when they were physically active, but stress, anxiety, sleeping disorders, and tiredness all decreased the motivation to exercise [53].

Of the 5484 migrants employed in Shanghai and interviewed in 2017 by Yang H, 21.1% had potential mental health problems, and 63.1% had an unhealthy lifestyle. The three most prevalent mental disorders were obsessions, compulsions, and interpersonal sensitivity and hostility. The female participants exhibited significantly increased mean scores for phobic anxiety and anxiety ($p < 0.001$) [54].

Finally, the variables associated with a decrease in psychological distress, found by Texeira and colleagues, are being a male, being satisfied with their income level, living with the core family and having higher number of children and planning to remain for longer periods in Portugal. Study variables negatively associated with immigrants' distress were job insecurity and the perception that health professionals were not willing to understand immigrants during a clinical interaction [55]. Rodriguez and colleagues, through their analysis, highlight that physical work demands were often accompanied by psychological work demands. A little over half of the women investigated reported low job control, especially in the service occupations. These workers received hostility rather than support from their supervisors and their coworkers. Some women reported some mood instabilities because of increased frustration from work, which resulted in stress and depression [56].

In contrast, some authors found no gender differences; for example, Yu B, among 1293 Chinese rural–urban migrants, scored significantly higher on Migration Stress ($p < 0.05$), somatization ($p < 0.05$), obsessive-compulsive symptoms ($p < 0.05$), depression ($p < 0.05$), anxiety ($p < 0.05$), and hostility ($p < 0.05$). Among female migrants, similar results were observed [57].

Similarly, Zhong in 2015 recruited and interviewed over 3000 migrant workers from 10 manufacturing factories. No significant gender or age-group differences were found. The risk factors for lifetime mental distress included lower education, worse living condition, poorer self-perceived physical health, migration before adulthood, and having done many jobs [58].

### 3.2.2. Alcohol and Nicotine Abuse

The data provided by the literature are conflicting. For many authors, the increased stress levels of migrant workers can lead to taking psychotropic substances, consuming alcohol and smoking.

For example, Arcury found that 48.5% of the farmworkers engaged in heavy episodic drinking in the previous three months, and 23.8% frequently engaged in drinking during this period. Significant factors for being at risk for alcohol dependence were stress and being a farmworker [59]. In the Gao's study, the overall prevalence of hazardous drinking was 10.6%, with male rural urban migrant workers having a statistically higher consumption of alcohol (18%) than females (2.2%). The rates differed by education level (elementary school had the lowest rate), by salary level, and by individual-level social capital [60].

Ismaylova and Gilbert had analyzed a random sample of 450 and 1342 migrant market workers in Kazakhstan, respectively. The first found that alcohol-related problems were more prevalent among older workers, men, with higher levels of education, and those who were unmarried [61]. In Gilbert's paper, 16.4% of the workers engaged in hazardous drinking, and 73 participants (5.5%) reported having experienced childhood sexual abuse. Hazardous drinking, higher levels of depression, and lower levels of social support were significantly associated with perpetration of partners' violence in the prior months [62].

Mora wanted to emphasize how depressive and anxious symptoms can lead to more alcohol or tobacco abuse; in fact, in this study among 370 farmworkers, 16.7% participants had substantial depressive symptoms (CES-D $\geq$ 10), 8.8% had substantial anxiety (PAI $\geq$ 27), and 50.1% had the potential for alcohol misuse (AUDIT-C $\geq$ 4). There was, however, a significant association between

high depression scores and alcohol misuse, 21% of those that had a CES-D score of 10 or greater also had a score of 4 or greater in the Audit test ($p = 0.02$) [63].

Zaller also highlights how depression is possible risk factor, by a cross-sectional survey of 358 young migrant women working in entertainment venues in China. 203 (57%) participants had an AUDIT score ≥ 8 (risky drinking) and 95 (27%) women had an AUDIT ≥ 16 (probable dependence). Greater likelihood of probable alcohol dependence was associated with being younger, working at an affluent venue, and depressive symptoms during the past week [64].

Steward surveyed 1085 migrant workers in two South Indian municipalities, investigating risky behavior, such as unprotected sex with female sex workers and alcohol abuse. In multivariate logistic regression, the expectancy of having more fun helped drive the combination of alcohol and unprotected sex with partners (OR = 1.22, $p < 0.05$). Men concerned about alcohol-induced deficits were less likely to drink with partners (OR = 0.81, $p < 0.01$) but more likely to have unprotected sex with them (OR = 1.78, $p < 0.01$) [65].

Other authors have not found particular increases in alcohol abuse among migrant workers. According to Kumparatana, the vast majority did not have an alcohol problem (91%) or clinical depression (94%) [66]. Similarly, in the Bacio's study, the number of drinking occasions, total drinks per drinking episode, and number of binge drinking episodes were low among a sample of Latino day laborers. The average AUDIT score was 6.53 (SD = 8.15), which falls below the suggested cut-off for hazardous drinking of 8 [67].

On the subject of tobacco dependence, Cui found that rural–urban migrant workers manifested a high prevalence of life stress and work stress, both forms of stress associating with smoking. Of the 1595 migrant worker participants in this study, 1030 were current smokers, with smoking prevalence at 4.9% [68].

In Wu's research among 4189 participants, the current smoking rate was 28.3%, the daily smoking rate 21.2%, and the occasional smoking rate 7.1%. The average number of cigarettes smoked daily was 15.82 (SD: 8.09). In both regression analyses, the adjusted odds ratio increase with age, occupation (migrants working in construction than others), length of migration and tend to vary inversely with education [69].

In Liu's paper, 5380 migrants in China completed the questionnaire, among whom 45.0% of male participants and 2.0% of female participants reported being current smokers. Multivariate analysis revealed smoking in female migrants to be significantly associated with working at construction (OR 8.08), hotels/restaurants (OR 5.06), and the entertainment sector (OR 6.79). The male migrants working at construction (OR 1.30), entertainment sector (OR 1.86), being divorced (OR 2.20), with duration of migration of four or more than four years (OR 1.42), and number of migratory cities of three or more than three (OR 1.42) showed an excess smoking prevalence [70].

Finally, in Mou' research, the prevalence of daily smoking was higher in men (27.3%) than women (0.7%). These rates are significantly lower than national smoking rates (59.5% in men, 3.7% in women). Longer working hours and less rest were associated with higher rates of smoking [71].

### 3.2.3. Trauma, Violence, and Suicidal Ideation

Sexual, physical, and verbal abuse were detected in foreign domestic workers in Lebanon by Zahreddine's research (12.5, 37.5, and 50.0%, respectively); 66.7% of them were diagnosed with brief psychotic episode, with the mean duration of hospital stay of 13.1 days. Striking phenomenological findings among domestics were acute anorexia (39.4%), nudity (30.3%), catatonic features (21.2%), and delusion of pregnancy (12.1%) [72].

Ayalon in 2012 studied 178 Filipino homecare workers; the majority of the sample was married (56.7%) and female (87.6%). Mean age of respondents was 37 (SD = 6.3), and the average number of years of education was 8.3 (SD = 5.0). The average length of stay in Israel was 5.5 (SD = 2.4). 35% of the sample reported exposure to some type of abuse within their home/work environment. The most common types of abuse were "asked to do more than one's job requirement" (35%), "been

shouted/sworn at" (30%), and "been told offensive stories or jokes" (20%). The most frequent exposure to everyday discrimination was "people think as if you are not smart" (mean = 2.05). The most common major lifetime discrimination was "been unfairly stopped, searched, physically abused, or threatened by the police" (10%). Abuse within the home/work environment was the only predictor of depressive symptoms, with greater abuse being associated with higher levels of depressive symptoms [73].

The same author conducted a survey of 245 Filipino homecare workers in 2009 in order to evaluate their working conditions, exposure to abuse, and their clinical correlates. Here, 41% reported being verbally abused, and 40% reported not receiving adequate food; almost half reported work-related injuries. The most consistent predictor of burnout was exposure to work-related abuse [74].

Cheung and colleagues also examined domestic workers' health; among 105 foreign domestic helpers surveyed, all of them were women. They were more likely to be aged 30–39 years (59.6%), married (51.5%), and worked and lived in private houses (38.7%). They had an average of 4.18 years of work experience. In the study, 20.5% and 34.4% had experienced physical and verbal abuse, respectively. However, 16.7% of the abuse victims did not report their cases to any individuals or organizations; among those who reported, only 19.4% reported their cases to formal organizations, including the employment agency and police. Although the majority of the respondents showed no depression, 25.2% had mild or severe depression level [75].

In Green's study, almost all the home care workers reported exposure to certain workers' rights violations; for example, 58% of them did not receive any vacation days besides the weekly day off, 30% reported not get even a weekly day off on a regular basis, and 79% did not get paid sick days.. A smaller portion (7.4%) reported work-related abuse. When compared to local workers, migrant home care workers were more vulnerable to some worker's rights violations, as well as emotional abuse [76].

Abuse, occupational hazards, physical and mental health, and post-trafficking well-being are described in a Pocock's paper among a sample of 275 trafficked fishermen in Thailand and Cambodia. Common physical health problems included dizzy spells (30.2%), exhaustion (29.5%), headaches (28.4%), and memory problems (24.0%); nearly one-third (29.1%) reported pain in three or more areas of their body, and one-quarter (26.9%) reported being in "poor" health. Physical health symptoms were strongly associated with severe violence, injuries, engagement in long-haul fishing, immigration detention, or symptoms of mental health disorders [77].

Meyer analyzed what migrants in some Asian countries, such as Thailand and Cambodia, suffer very often. Almost all 33 respondents mentioned problems with the police in Thailand, including being arrested, forced to pay a bribe to be let out of prison, and being deported back to Cambodia by the police. The central problems were related to poverty (*n* = 30), including debt (*n* = 28), landlessness (*n* = 16), not having enough money to send children to school (*n* = 10), and not having enough to eat (*n* = 16). Other mental health issues, including alcohol abuse (*n* = 6) and domestic violence (*n* = 9), were commonly reported [78].

Chen G has analyzed the prevalence of "post-traumatic disorders" among migrant workers in an area of China that results 17.42%. Being male, being older than 35 years, being an electrician, having dependent children, suffering property damage, being without medical insurance, and having low social support were risk factors significantly related to the development of this disorder [79].

Two hundred workers from four bazaars and 200 workers from 18 construction sites completed a Weine's survey of Tajik married male seasonal labor migrants in Moscow. The average age was 31.5 years (SD = 6.1). Education level was 6% primary, 72% secondary, 9% college, and 10% university degree. All men were married to a woman in Tajikistan, and they had an average of 1.9 (SD = 1.3) children in Tajikistan. The five most experienced direct trauma exposure factors were beaten by the police (45%), had health problems without access to medical care (19%), were beaten by nationalists (17%), lack of food or water (14%) lack of shelter (14%). 34% of migrants reported one or two direct cases of traumatic cases; these subjects reported nightmares (32%) and be careful, attentive, or scared

(34%). Migrants who tested positive for PTSD were younger, less educated, came from different regions of Tajikistan, lived with more people due to the economic crisis, and had worse health [80].

Espinoza-Castro's cross-sectional study included a sample of 282 Latin American migrants living in Germany; 45% of the study population reported symptoms of distress, and 63% of the population worked below skill level. The 12-months prevalence of violence at the workplace was 14%. After adjustment, working below skill level was related statistically significantly to distress (OR 2.80) [81].

Finally, in an interesting article by Heeren that compares various categories of migrants, it was reported that asylum seekers reported more experienced traumatic event types than illegal migrant, labor migrant, and residents; they showed a much higher rate of probable PTSD diagnoses and clinically relevant symptoms of depression than all other samples. Of all socio-demographic, pre- and post-migration factors assessed, only social desirability, post-migration resources, potentially traumatic events, and resident status were associated with mental health outcome [82].

As for the extreme attempts to take one's own life, authors have often found a correlation with depressive symptoms. Through Luo's research, over 15% among 5115 unmarried female migrant workers during 2015 to 2016 from Shanghai experienced induced abortion and over 8% reported suicidal ideation during the past year, especially among those with a lower education and a shorter stay in the workplace, more likely to drink alcohol and report daily internet use. The association was strongest in those who had 25 years (OR—3.37), with 5 years of stay in the workplace, OR—2.98), the non-anxiety group (OR—2.28), and the non-depression group (OR—2.94) [83].

Al-Maskari conducted a cross-sectional survey among 200 laborers in Al Ain City's labor camps. The researchers found that depression was correlated with physical illness, working in construction industry, earning less than 1000 UAE Dirham per month, and working more than 8 h a day; 6.3% of the study participants reported thoughts of suicide, and 2.5% had attempted suicide. People with suicidal ideation were more likely to have a physical illness, earn less than 1000 UAE Dirham per month, and work for more than 8 h a day [84].

Warfa studied Somali professionals living in UK and USA. Of the 189 respondents, 52% were men and 48% were women. The prevalence rates of current major depression, suicide ideation and current agoraphobia were higher among the UK respondents compared with the USA respondents (27% and 7%, respectly). Thwarted aspirations, devalued refugee identity, unemployment, legal uncertainties, and longer duration of stay in the host country account for poor psychological well-being and psychiatric disorders among this group [85].

Finally, according to Habtamu's paper, the prevalence of common mental disorders among migrant returnees was found to be 27.6%. The authors found the following symptoms to be highly prevalent: headaches (40.6%), poor appetite (39.4%), fatigue (35.8%), difficulty sleeping (36.9%), feeling unhappy (37.6%), feeling nervous or tense (32%), hand tremors (14.6%), trouble thinking clearly (19.3%), suicidal ideation (15%), problems with decision making (20%), and functional impairment (19.9%) [86].

### 3.2.4. Sleep Disorders

Grzywacz, in 2011, described depressive symptoms and daytime sleepiness among immigrant Latino farmworkers across the agricultural season; approximately 40% of the sample were 40 years of age or older, another 35% were 30–39 years of age, and the remainder were between 18 and 29. Half of the sample (*n* = 62) had six years or less of formal education, while another 40% reported having between seven and nine years of education. He outlined associations of depressive symptoms with drowsiness. Results indicated that 45% of agricultural bracelets experienced high depressive symptoms throughout the season, while 20% experienced them during periods with higher workload. Drowsiness was more common among women than among men and it can cause an increase in injury rates. [87].

Sandberg exposed the problem of poor sleep quality among migrant workers through several studies: in 2012, from a cross-sectional survey of 30 migrant farmworkers conducted in North Carolina,

he found that 11% of migrants reported elevated levels of daytime sleepiness, 28% reported elevated levels of depressive symptoms, and 5% reported moderate to severe musculoskeletal pain [88]. In 2014, he found that poor sleep quality was positively and significantly correlated with length of time falling asleep ($p < 0.001$), the number of night awakenings ($p < 0.001$), and negatively associated with hours slept ($p < 0.001$). Sleep quality was significantly associated with air conditioning use. In fact, 45% of farmworkers who had indicated they had good sleep quality reported having air conditioning in the housing unit [89]. Finally, in 2016 he reported that sleep quality can be influenced by long hours of work (61% worked more than 40 hours/weekly), pesticide exposure (those reporting moderate pesticide exposure were least likely to report poor sleep quality), and elevated depressive symptoms ($p < 0.001$) [90].

### 3.2.5. Work-Related Stress

Work-related stress involves various factors such as any inadequacy in the management and organization of work processes, working conditions, and environment, communication, and personal factors. Many researchers found that workers with perceived high levels of work demands were more likely to suffer these psychological disorders.

For example, Lee H analyzed 200 Korean-Chinese full-time migrant workers; roughly 30% of the sample met the criteria for depression, especially women; there were moderate positive correlations between depression and job demand ($p < 0.001$), insufficient job control ($p = 0.003$), interpersonal conflict ($p = 0.001$), and acculturative stress ($p < 0.001$) [91].

Migrant workers recruited by Capasso with perceptions of high-work demands and work stress were more likely to suffer anxious-depressive disorders; 60.6% ($n = 272$) and 62.1% ($n = 278$) of migrant workers with high perception of work demands reported respectively high levels of anxious-depressive disorders and high levels of interpersonal disorders, while 65.0% of those with high perception of rewards reported low levels of interpersonal disorders [92].

Vanhoutte compared English migrants to Australian natives; the first group reported more heavy labor or physically demanding work ($p = 0.035$), workplace danger or injury risk ($p = 0.002$), leaving the job because of ill health or disability ($p = 0.002$), or feeling that their health had been affected by their job ($p < 0.001$) [93].

Sixteen respondents in Zhong's study in 2016 said that they had to work overtime, 27–28 days per month, and 12–14 h per day [94]. A work environment with latent health hazards was identified as another main source of work-related stress for all of the 17 migrant workers; the night shift was further recognized as complicating their daily lives, and 10 participants said that their supervisors were very harsh on workers and even strictly controlled workers' time spent in the washroom.

In Subedi's search, 146 participants spent 5.3 years in their current occupation and worked 45.5 hours per week, which is considerably higher than the 36.6 h averaged by Canadians; 35% of the participants were dissatisfied with their work and 12% were highly dissatisfied. Working only at night increases the likelihood of reporting ill health (9.54 times greater than working in the morning alone), as well as a high level of stress (1.6 times greater) [95]. Subedi itself, in 2014, tested four different hypotheses (lifestyle change, barriers to health services, poor social determinants of health and work-related stress) in order to understand changes in the health status of migrants. The study concluded that there is a statistically significant difference in the socioeconomic characteristics and health outcomes of immigrants who are less than 10 years of residence in Canada [96].

Yao C.'s results among 88 Chinese workers in Italy show that organizational factors are significantly and positively related to emotional well-being at work, occupational health and performance ($p < 0.01$). Further analyses also show that certain personal factors, such as education level, language skills and new social networks can have a moderating effect on the perceived relationship between organisation and stress in the workplace [97].

Li W's study confirm this assumption; in fact, in this analysis, higher work stress scores were correlated with higher depressive symptoms, especially if female, having poorer self-rated health,

and having a lower self-rated social class. Among the moderators, community supportive network and community cohesion were related to fewer depressive symptoms; community supportive network moderated the relation between work stress and depressive symptoms (B = −0.281, *p* < 0.01) [98].

Other authors have investigated the degree of job satisfaction; 54% of housekeeper participants interviewed by Hsieh indicated that they were satisfied with their jobs, 23% expressed mediocre satisfaction and 23% were not satisfied with their jobs due to reasons such as heavy physical work, low pay, and discrimination by supervisors and co-workers. About half of the interviewees felt that their jobs were secure and that they could keep their jobs as long as they wanted, while the remaining 46% felt that their jobs were insecure and that they sometimes worried about losing them. 15% of housekeepers reports high stress levels from work so manifest to require medical treatment [99].

In Nilvaranguk's article, migrant workers often have to accept any offer of paid work and consider themselves lucky to find paid work in Thailand; some expressed dissatisfaction due to low pay, unhealthy workplaces, many accidents and physical/verbal abuse [100]. A cross-sectional study was implemented by Nguyen's survey of 2818 female workers in 10 light manufacturing factories; only a few female workers (2.8%) said their income can meet family needs, while 14.4% of women said their salary is not enough to cover their day-to-day expenses. More than 28% of workers who had to rent accommodation claimed that their income was not even enough to cover this expenditure (18.5%) [101].

Linguistic and cultural elements may play role in the stress reported by migrants, because can isolate the worker from the context and cause depressive symptoms; consequently, isolation can lower the quality of work performance of the subjects involved. In fact, Rosano found that language requirements create a barrier to migrant workers obtaining better employment—in Italy and Spain, the risk of stress was predominant among natives (in Italy OR = 0.30; in Spain OR = 0.86) [102]. Bousman confirm that neuropsychological impairments can lead to difficulties in learning new jobs, in dealing with employers and colleagues, in daily functioning, poor adherence to both medical and behavioral interventions and risky health behaviors (e.g., cigarette consumption) [103]. Differences in language is a main stressor which impedes team collaboration. In fact, in a Schilgen's analysis among migrant nurses, they perceive time pressure, lifting patients, lack of appreciation, or the client's personal fate as burdening [104].

The divergent understanding of behavioral patterns as well as of nursing care and a non-functioning communication impede the collaboration within a diverse nursing workforce. Migrant and minority nurses suffer prejudices, verbal and sexual harassment proceeding from their clients.

Anjara conducted a cross-sectional survey of 182 domestic migrant women in Singapore; although women reported a relatively good overall quality of life, more than half of the participants reported feeling stressed. In addition, almost 20% of the participants reported feeling very isolated from the social context; the social connection and proper management of the organization in the workplace have been positively associated with the quality of life, improving the state of physical and mental health. [105].

Longer length of migration and poor social support were significantly associated with poor Work Ability Index. Han found that age, marital status, and better social support were associated with higher WAI grade; worse social support, younger age, worse physical environment, lack of work control, and high work pressure were all significantly related to weak perceived work ability [106]. Increased isolation and poor social integration may especially be seen in workers who undergo major life changes, such as transferring from a rural to an urban zone and moving away from family members. This data is reported in some research. Zhu C in 2012 analyzed 3622 rural–urban female migrant workers. These were facing poor social support, social accommodation, living and work environment; as a result, they had lower quality of life compared with the general population [107].

Work duration showed a negative association with role physical, general health, social functioning, role-emotional and mental health. Simsek's study consisted of 270 randomly selected seasonal migrant agricultural workers. The risk factors of stress emerged were workplace physical conditions (25.7%),

workplace psychosocial, economic factors (19.3%), workplace health problems (15.2%) and school problems (10.1%) [108].

In 2012, rural urban migrant workers analyzed by Yang T showed a higher prevalence of mental disorders (24.4%), higher than that of urban residents (20.2%). For the genesis of these symptoms, some risk factors were identified, including being from the South (OR 2.00), increased life stress (OR 7.63), staying in the city for 5-9 months per year (OR 2.56), increased work stress (OR 2.56), separation from wife (OR 2.43) and be employed in the transport sector (OR 0.54) [109].

Work related stress manifests itself on the bodies with somatization and physical problems for the workers; most of the women employees analyzed by Akter said that they lived with constant feelings of guilt, anxiety, poor appetite, and that these feelings contributed adversely to their mental well-being. They felt restless, fatigued, and wanted to die due also to longing for their children [110].

Winkelman interviewed 57 farmworkers in Carolina. They often felt tired from standing all day at work; they had tension in their head and neck; others reported that they were not given breaks to alleviate being tired and that they worked too many hours per day [111].

Tsai and colleague found that 156 white-collar migrant workers reported a high prevalence of perceived work-related stress (62.2%) and a lower prevalence of regular exercise (12.2%); workers with higher levels of perceived work-related stress reported more alcohol consumption, a history of hyperlipidemia and a higher prevalence of self-reported neck pain, poor sleep, and mild/moderate/severe depression [112].

### 3.2.6. Poor Mental Health

Work is an important determinant of health and, in particular for positive mental health. Several authors have analyzed the quality of life of migrant workers in relation to working conditions. For example, the 2008 international economic crisis has contributed to worsening the health status of workers, starting with the weakest categories.

Agudelo and Robert interviewed many migrant workers to assess changes in mental health, after the eruption of the economic crisis in Spain. The first author found that the likelihood of suffering from poor mental health was higher in men, the unemployed, low wages, and those reporting household burdens [113]. Robert and colleagues also found an increased risk of poor mental health in the unemployed (OR 3.62), in workers at a tight pace (OR 2.35) and with lower incomes (OR 2.75) [114].

Chen, who found that the majority of the unemployed migrant workers were young males, in manufacturing industry, with low education level, also studied the impact of the economic crisis on the mental state of workers. Nearly 50% of unemployed migrant workers were classified as mentally unhealthy, and the most frequently reported symptom was depression; compared with the adult norm of 1986, 2003, and 2007 in China, unemployed migrants had more mental problems. Migrant workers who had been unemployed for a long time had more psychiatric symptoms than those who had recently lost their jobs; unemployed migrant workers with poor coping strategies and difficult adaptation obviously expressed many more symptoms than workers with suitable coping strategies [115].

In Zeng's research, about 35.3% of the sample examined had poor psychological well-being: in particular, they were men, younger in age, with longer weekly work hours, more exposure to hazardous work environment, higher job demands, and lower job autonomy, which were all significantly associated with worse psychological well-being [116].

Cayuela and colleagues found that settled immigrant women have a higher prevalence of poor self-perceived health (34.6%) and poor mental health (30.1%) than native women (17.7%). The most influential factor in the relationship between health and migrant status for women workers was occupational social class (25% for poor self-perceived health and 17.6% for mental health) and job satisfaction accounted for 15.8% of the difference in self-perceived health [117].

The Yang's study highlights how the category and workload can affect mental health; in this cross-sectional study of 5484 rural migrants who had worked in Shanghai, 11.2% reported abnormal mental health, correlated with unhealthy lifestyle. Men working in manufacturing reported less

unhealthy lifestyle than those in hospitality or recreation/leisure; women working in manufacturing and construction reported less unhealthy lifestyle than those in all other sectors. Unhealthy lifestyle was associated with small workplaces for men, working more than eight or 11 h per day for women and men, respectively [118]. The results founded by Kaewanuchit highlight that job conditions and distance travelled between house and workplace had a direct effect on mental health with a standardized regression weight of 0.581 and 0.443, respectively ($p < 0.01$). It was found that housing conditions had no effect on mental health, conversely the income variable had a direct influence on mental health with a standardized regression weight of 0.68 ($p = 0.01$) [119].

Furthermore, the scarce perception of safety in the workplace, due to a lack of staff training, few protection devices, and therefore an increased risk of accidents can lead to a state of discomfort and tension.

Exposure to occupational hazards showed negative influence on psychological health, which was supposed to be an influential intervening predictor. In fact, in research of Zhang, exposure to occupational hazards showed negative influence on psychological health in the model of total population ($p < 0.01$) and of age $\leq 45$ model ($p < 0.01$) [120].

Socially isolated migrant and seasonal farmworkers are particularly susceptible to the effects of natural disasters. The Burke's research project assesses the awareness, perceived risk and practices regarding disaster preparedness and response resources and identified barriers to utilization of community and government services during or after a natural disaster among Latino migrant workers. Only 37% of the group had received information on how to prepare for a natural disaster [121]. In Daly's study, authors found disparities in the prevalence of working in jobs with adverse psychosocial factors between Australian-born and migrant workers; overall job adversity, working in complex or demanding jobs, and low security were associated with probable mental health problems in all workers [122].

Social support reduced the impact of acculturative stress and migrants with support are more likely to access health care. Holumyong's study utilized data of 987 migrant workers in Thailand, and the results confirmed the importance of acculturative stress and social support among this group of migrants; the language barrier, a crucial factor behind acculturative stress, adversely influenced access to maternal care [123].

In Ang's survey of 433 non-domestic migrant workers, 61.4% of workers said they had insurance, but did not fully understand all the benefits. In fact, most workers are often unsure whether they have received the correct information in this regard (72.4%), often also due to language barriers (67.7%) [124].

All Latino farmworkers interviewed in Viveros-Guzman's article spoke of difficulties in the realm of communications with supervisors; on some farms, supervisors were also foreigners whose first language was neither English nor Spanish. Miscommunications and stress are common outcomes [125]. Li J found that social exclusion is negatively associated with migrants' mental health: limited access to full labor rights and experience of social stigma, discrimination, and inequity were the most significant factors [126].

## 3.3. Trial

The aim of Zhu's trial was to determine the effectiveness of an educational program on 3344 Chinese female migrant factory workers' quality of life and job satisfaction. During the intervention period, the female migrant workers received both the usual community-based health services and the intervention package, consistent in educational materials and lectures about reproductive health, occupational health, lifestyle and behavioral interventions (physical activity, dietary habits, personal hygiene, sexual behaviors and occupational protection). 84.0% were single, 82.0% were less than 25 years old, 74.0% received only elementary education and 70.0% participants were in the first year of employment. The participants reported higher General Health, Vitality, Social Functioning, Mental Health scores (all ps < 0.05) at the six-month survey and they reported higher job satisfaction (25.8% vs. 13.4%; $p < 0.001$) [127].

The data used for Mc Coy's study was drawn from a trial of alcohol risk reduction intervention project, targeting migrant workers recruited in an agricultural area in Florida. The sample was composed of 431 participants, number reduced to about 271 at 12-month follow-up. The study assessed the effectiveness of a cognitive behavioral program called "Peer Education Ends Risky Behaviors" (PEER) in comparison with a health promotion program "Health Education Always Leads to a Healthy You" (HEALTHY), to which the participants were randomly assigned. The sample at baseline was composed of 203 male migrant workers, 77% Hispanics, 33% Africans, 82.8% single with an average age of 41.45 years. African Americans reported significant decrease in the number of drinks in last 30 days ($p = 0.024$) and number of drinking days in past week ($p = 0.021$) from baseline to 12 months, while Hispanics reported decrease only in the number of drinks in last 30 days ($p = 0.045$). Among Hispanics, all psychosocial variables showed a significant change in social support ($p = 0.023$), behavioral intention ($p = 0.051$), self-efficacy ($p = 0.023$). Instead, among African Americans, there was a significant increase only for the top 2 factors [128].

Finally, Smith's experimental trial evaluates the impact of a multi-component intervention on the psychosocial work environment at a multi-ethnic Danish workplace in the cleaning sector. Ten women and four men participated in the intervention. They were employed in public schools and originated from six different countries including Romania, Thailand, Turkey, Serbia, Philippines, and Denmark. Six of the immigrant cleaners had lived and worked in Denmark for less than two years (newcomers), and six had been in Denmark for more than 10 years (long-term residents). The intervention included Danish lessons, vocational training courses, a workshop on job satisfaction and teamwork for all the cleaners. Individual semi-structured interviews were conducted with the cleaners at their work place three weeks before and three weeks after the intervention. After the intervention, most of the cleaners expressed that the communication and collaboration internally in the group of cleaners had improved; in fact, most of the cleaners expressed that improved language proficiency and increased interaction with colleagues had led to improved communication, trust and collaboration. Positive changes were observed concerning COPSOQ scales such as meaning of work, perceived rewards, level of emotional demands, social support from supervisor and colleagues as well as job satisfaction, social community and trust regarding management [129].

## 4. Discussion

According to the latest "Report on the Health of Refugees and Migrants", foreign workers constitute the largest group globally [130].

In 2015, the International Labour Organization (ILO) registered about 60 million migrant workers in Europe (12% of the entire working population) [131]. The health of the migrant worker has therefore become a priority for the European and World Communities. Labor migrants reported musculoskeletal, respiratory and mental health disorders. According to our results, migrant workers develop more anxiety-depressive symptoms than resident ones. Personal history of the migrants (e.g., past physical illness, infrequently visiting hometown), their socio-economic context (e.g., poor living condition), and their work environment (e.g., low monthly income, having worked in many cities, working more than 8 h/day) therefore greatly influence the development of mental health problems [132].

The personal characteristics of the migrant workers, such as the gender, can increase the risk of developing mental problems. While work-related injuries are more common in male migrants due to heavier and more dangerous jobs, regarding mental health, women are more stressed because of their distance from the family, caregiver jobs, and the greater frequency of harassment [133]. Migrant workers often leave home to ensure greater economic income for their families. This choice, while improving the economic conditions of relatives, has negative effects on the psychology of the migrant [134]. The distance from home and the lack of family support causes to the migrant worker a growing sense of insecurity and loneliness. The numerous tools available (e.g., social network) to maintain contact with family members do not help the sense of isolation of the migrants, who often develop anxiety-depressive syndromes.

Preventive strategies are necessary. Filipino domestic workers, for example, reduce migration-related stress by maintaining cultural identity (e.g., praying, reading the Bible) [135].

The new socio-economic context also influences mental equilibrium. Migration, difficult adaptation, different lifestyles and languages generates stress in workers [136]. Migrants, due to poor integration, have greater barriers to access to health services and therefore are unable to manage stress-related problems (e.g., insomnia, gastrointestinal symptoms, headaches).

Dick et al. evaluated the screening of the emotional distress (ED) in the patients of the public clinic of Physicians for Human Right Israel, which provides free health services to migrants without health insurance. The results showed an underestimation of migrants' ED by general practitioners (GPs). One of the causes could be a "nihilistic approach" of general practitioners who consider mental health problems to be naturally more frequent in migrant workers due to their condition. Health professionals therefore need specific training for migrant management to give them the best possible assistance and avoid discrimination [137]. Difficult access to health care is even more important in migrants with basic chronic conditions, such as schizophrenia. In China, through a registration system (*Hukou*), migrant workers are classified as temporary residents of the city where they work, with the consequent disadvantage when receiving social assistance (e.g., health services, public schools, subsidies for housing, occupational safety).

Because of poor access to healthcare, migrant workers with schizophrenia prolong their duration of untreated psychosis and thus increase their need for hospital care [138].

The working environment, together with the distance from the family and the new culture, can drastically worsen the migrant's mental health status. Cieslik defines "*Quality of employment*" as "*the conditions in which work takes place, i.e., possibilities for professional development and creativity, stress levels, workplace relations and job security*" [139].

Poor "quality of employment" causes negative consequences on the migrant employees such as work-related stress [140]. Our data showed that migrant worker are more often affected by work-related stress than residents are. A cause of stress is the disparity between the skills of the migrant worker and the characteristics of the job. This condition is called "*under-employment*" and is used when a job is lacking in relation to professional skills. In Australia, there is a strong under-employment among recent migrants. After about three years in the new country, migrant workers do jobs for which they are over-qualified and this causes a lot of symptoms such as anger, irritability and emotional distress [141]. In Europe, under-employment conditions have been reported among migrant workers. For example, in Finland, migrant doctors are more often employed in the primary care sector because difficult recruitment from the native workforce. Migrant physicians develop burnout due to the lack of professional support [142]. Even migrant workers with less qualifications and a lower socio-economic status, such as Chinese rural–urban workers, develop burnout due to chronic exposure to work-related stress [143].

The mental status influences both the country of departure (financial help for the family) and the country of arrival (socio-economic development); it is therefore one of the main objectives of the WHO for the maintenance of global health. Preventive strategies for the protection of the mental health of the migrant worker are necessary. A multilevel systemic approach based on the personal characteristics of the migrant worker is indicated for prevention of anxiety-depressive symptoms [144]. In Italy, the "Trans-cultural Psychosomatic Team" was created to provide health care to migrant workers with basic pathologies (e.g., schizophrenia) [145].

Work is also an important factor for the integration of the migrant. Favorable working conditions facilitate the inclusion of migrant workers and avoid discrimination [146]. Therefore, migration, if based on health and work assistance policies, can become a factor promoting the health and well-being of the individual [147,148]. For example, interventions to improve the psychological health of the employees should be introduced; in particular, it is necessary to improve employees' motivation, provide feedback and increase employee participation [149]. Interventions are aimed at building strengths, enhancing positive individual resources [150,151], promoting well-being [152,153].

This systematic review has some limitations. First, in our analysis there are only three trials, which are fundamental for understanding the determinants of occupational diseases and founding new appropriate interventions.

Studies are often conducted outside the European context, in small groups of workers, in selected groups and often the results from migrant workers have been compared with the data of the natives, which does allow for the adequate characterization of risk. Overall, the quality of the studies was not too high, due to the frequent use of subjective assessment tools, such as questionnaires not always standardized, poor description of the sample'characteristics and of the subjects possibly excluded.

Finally, it was very difficult to underline common points and differences of the various studies because they differed significantly by type of sector of work, social contexts (such as culture, religion, legislation) and involved countries.

## 5. Conclusions

Considering the constantly growing trend of migratory flows and the particular susceptibility of this vulnerable group, doctors must direct their attention and their commitment to the protection of this category through the available scientific data and appropriate preventive interventions. To deal with the impact of radical change and improve the integration process, occupational physicians and entrepreneurs will increasingly have to consider several factors. It is essential to find out the predisposing factors for the development of mood disorders during medical examinations, as well as the personality of the migrant worker and the level of individual empathy. It is also important to identify the coping strategies implemented by the workers, through which the whole team can contribute to reduce the negative impact of change, stress, psychosomatic disorders, and decreases in productivity. It will be necessary to prepare the entire working staff for interaction with the various cultures and for developing the necessary openness to acquire new knowledge, develop a critical spirit, and create an intercultural working environment through individual initiatives and targeted actions. Companies can increase this cultural exchange through the creation of networks, opportunities for dialogue, encouraging foreign workers to ask questions, and organizing debates. These initiatives could take place during specific training days or meetings in common places, such as the company canteen, to reduce the greater tendency of foreign workers to isolate themselves.

**Author Contributions:** Conceptualization, N.M., V.T., and G.G.; methodology, N.M., V.T. and G.G.; formal analysis, N.M., V.T., and G.G.; writing—original draft preparation, N.M., V.T., and E.T.; writing—review and editing, G.G., M.C., and S.D.S.; linguistic revision: E.T.; supervision, N.M. and G.A.; project administration, G.A. All authors have read and agreed to the published version of the manuscript.

**Funding:** This research received no external funding.

**Conflicts of Interest:** The authors declare no conflict of interest.

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
