# Peer review of "Migrant Workers and Psychological Health: A Systematic Review"

_sustainability, doi:10.3390/su12010120_

Round 1
Reviewer 1 Report
The present study presents a systematic review of the empirical literature relating on mental health of migrant workers. The review included 127 studies published from 2009 to 2019 with references from three major online databases—Pubmed, Cochrane Library and Scopus. The authors identified several risk factors for migrant workers’ mental health: substance abuse, work-related stress, abuse at work, both physical and verbal, limited access to mental health services, in particular, and to health care system, in general. The authors address an important topic, and it appears to me that this study is a part of a larger project. The manuscript is worthy of publication in Sustainability and will contribute to academic knowledge on migrant workers’ health. However, the Conclusion section is not clear to me and should be extended. The authors should mention implications of their findings for future research, as well as the plausible innovative public health approaches to the problem of migrant workers’ mental health.  Â
Author Response
Dear reviewer, we thank you for your corrections. We have broadened our conclusions by trying to better explain the contribution of occupational medicine to migrant workers. We remain available for further advices.
Reviewer 2 Report
The article deals very carefully with the issue of the quality of life of migrant workers.
The literature is very extensive and the method used to analyse and summarise it is clearly described.
The results are significant and presented in a coherent way.
I only point out the need to make tables 1, 2, 3 more readable, these tables are very important for the understanding of the research process.
Â
Author Response
Dear reviewer, we thank you for your corrections. We have made the tables more readable and better understtod. We remain available for further corrections.
Reviewer 3 Report
I realize that great work and time has been devoted to this paper. It has a lot of strengths, but I think that some changes should be recommended.
First, the title does not orient the readers well about the content of the paper: the title talks about the Psychological health phenomenon, but the paper is focused on the migrant population, so I would suggest to re-write the title to put migrant workers as the first because it is the newest and different from existing research on health.
Second, less information appears in the abstract. Maybe expanded by adding the most relevant findings.
The theoretical review is complete and updated, the objectives of the research have been clearly defined, the methodology for the selection of the sample of studies and the data collection is adequate, the applied statistical analysis answers the questions of the investigation.
Related to the results, please, try to better describe the sociodemographic data of the primary studies' participants. In the same sense, give the readers with detailed information about the procedure for recruiting participants and collecting data in the primary studies.
Discussion:
First of all, try to better adjust your conclusions to the findings. Or to say in other words, please try to justify more clearly the connection between your conclusions and your findings.
Finally, a section related to limitations, future lines of investigations and the principal contributions of the research could be interesting. Your paper has a lot of relevant implications for society and policy – makers, but you need to elaborate more on this topic.
Conclusion:
They don’t appear new conclusions on this part. This part does not add any new to the rest of the paper. Please, try to condense your findings, or to highlight your main contribution to the field.
Â
Â
Author Response
Dear reviewer, we thank you for your corrections.
1) We have re-write the title to put migrant workers as the first; we opted for "Migrant workers and psychological health: a systematic review".
2) We expanded abstract by adding the most relevant findings.
3) we tried to better describe, in cases where it has been possible, the sociodemographic data of the primary studies' participants and the procedure for recruiting participants, entering the data in the text.
4) as suggested by the second reviewer, we have tried to better explain in our conclusions the role of occupational medicine and the contribution it can make in the face of alarming results.
we remain available for further corrections.
Thank you